# The Efficacy of Common Household Cleaning Agents for SARS-CoV-2 Infection Control

**DOI:** 10.3390/v14040715

**Published:** 2022-03-29

**Authors:** Catarina F. Almeida, Damian F. J. Purcell, Dale I. Godfrey, Julie L. McAuley

**Affiliations:** 1Department of Microbiology and Immunology, University of Melbourne at the Peter Doherty Institute for Infection and Immunity, Melbourne, VIC 3000, Australia; cdos@unimelb.edu.au (C.F.A.); dfjp@unimelb.edu.au (D.F.J.P.); godfrey@unimelb.edu.au (D.I.G.); 2Global Virus Network Center of Excellence at the Peter Doherty Institute for Infection and Immunity, Melbourne, VIC 3000, Australia

**Keywords:** COVID-19, pandemic, SARS-CoV-2, disinfectant, virucidal, antiseptic

## Abstract

The COVID-19 pandemic caused by SARS-CoV-2 is having devastating effects on a global scale. Since common household disinfectants are often used to minimise the risk of infection in the home and work environment, we investigated the ability of some of these products to inactivate the virus. We tested generic brands of vinegar, bleach, and dishwashing detergent, as well as laboratory-grade acetic acid, sodium hypochlorite, and ethanol. Assays were conducted at room temperature (18–20 °C, 40% relative humidity), and two time points were used to reflect a quick wipe (30 s) and a brief soak (5 min). Vinegar, and its active ingredient, acetic acid, were completely ineffective at virus inactivation even when exposed to the virus at 90% *v/v* (a final concentration equivalent to 3.6% *v/v* acetic acid). In contrast, ethanol was capable of inactivating the virus at dilutions as low as 40% *v/v*. Dishwashing detergent effectively rendered SARS-CoV-2 inactive when diluted 100-fold (1% *v/v*). Bleach was found to be fully effective against SARS-CoV-2 at 0.21 g/L sodium hypochlorite after a 30 s exposure (1/200 dilution of commercial product). Given reports of infectious virus recovered from the surface of frozen packaging, we tested the persistence of infectiousness after multiple freeze-thaw cycles and found no change in infectious SARS-CoV-2 titre after seven freeze-thaw cycles. These results should help inform readers of how to effectively disinfect surfaces and objects that have potentially been contaminated with SARS-CoV-2 using common household chemicals.

## 1. Introduction

The current Coronavirus infectious disease 2019 (COVID-19) pandemic, caused by the Severe Acute Respiratory Syndrome Coronavirus-2 (SARS-CoV-2), induces severe pneumonia in approximately 15% of unvaccinated patients, with fatality rates of more than 20% in the elderly and 8–12% of patients with comorbidity [1,2,3,4,5]. Vaccinations against SARS-CoV-2 have been effective in reducing COVID-19 severity [6], but the virus is still transmitting among the human population. This continued transmission has meant host-adaptive evolution of the virus leading to isolates of increased transmissibility, virulence and pathogenicity (coined variants of concern (VOC)). There is a valid fear that the VOCs will evolve mutations that render the antibody response in vaccinated individuals less effective and upon infection will result in increased viral load, COVID-19 severity, and prolong the periods of transmissibility, perpetuating the pandemic and the global economic burden [7,8].

In an attempt to control SARS-CoV-2 transmission, the Centers for Disease Control recommend exposed individuals isolate, or, if unable to do so, wear a well-fitting mask to minimize the risk of infecting people they encounter [9]. While this measure limits aerosol transmission risk, infected persons, whether they are symptomatic or asymptomatic, can release SARS-CoV-2 into the environment through respiratory expectorates via sneezing, coughing and skin contact, resulting in potential fomite contamination of surrounding surfaces [10].

Laboratory-based studies have demonstrated the persistence of infectious SARS-CoV-2 on surfaces [11,12,13]. However, difficulties in sampling infectious SARS-CoV-2 from environmental surfaces [14,15,16,17] has impacted the assessment of the likelihood of fomite-mediated transmission, and there is an ongoing debate about the degree of risk that infection by this route poses [18,19,20,21,22,23,24,25,26]. Regardless, numerous studies have focused upon the improvement of microbiocidal technologies, including testing of novel and existing antiviral surface coatings [19,27,28] and disinfectants [29,30,31,32,33,34,35] with the goal to reduce surface contamination and risk of SARS-CoV-2 fomite transmission. In addition to promoting increased hand hygiene and mask-wearing, many authorities have mandated interventions such as deep cleaning of surfaces at exposure sites, including schools, public transport, hospitals and hotel rooms, in order to reduce the risk of further potential transmission events [36,37,38]. Therefore, SARS-CoV-2 fomite contamination at sites with high touch surfaces remains a public health concern.

A survey by the CDC revealed that more than 80% of respondents did not feel they knew how to disinfect their home and clean safely to prevent SARS-CoV-2 transmission, despite increasing their household cleaning efforts [39]. To address this issue, we tested the use of readily available household cleaning products: vinegar, bleach and dishwashing detergent, as well as ethanol to represent alcohols available in the house, for the ability to render SARS-CoV-2 non-infectious. We investigated a range of dilutions of these products used for either 30 s (based on WHO recommendations [40]) or 5 min and provided a guide as to what the minimum dilution is for each product to still retain antiviral activity. Our results indicate that bleach, dishwashing liquid and alcohol-containing solutions, but not vinegar, are common household cleaning products that can be used as effective virucidal agents against SARS-CoV-2. Given out data, to decontaminate common household and workplace surfaces to reduce the risk of SARS-CoV-2, we recommend that these products are not diluted beyond a certain point that we define in this study.

## 2. Materials and Methods

**Cell Line Maintenance**: Vero cells (American Type Culture Collection ATCC) were maintained in Minimal Essential Media (MEM) supplemented with 10% heat-inactivated foetal bovine serum (FBS), 10 μM HEPES, 2 mM glutamine and antibiotics. Cell cultures were maintained at 37 °C in a 5% CO_2_ incubator.

**Virus stock propagation**: All virus infection cultures were conducted within the High Containment Facilities in a PC3 laboratory at the Doherty Institute. To generate stocks of SARS-CoV-2, confluent Vero cell monolayers were washed once with MEM without FBS (infection media) then infected with a known amount of the original cultured clinical SARS-CoV-2 isolate (hCoV-19/Australia/VIC01/2020) [41]. After 1 h incubation at 37 °C in a 5% CO_2_ incubator to enable virus binding, infection media containing 1 μg/mL TPCK-trypsin was added, and flasks were returned to the incubator. After 3 days incubation and microscopic confirmation of widespread cytopathic effect (CPE), the supernatants were harvested, and filter sterilized through a 0.45 μm syringe filter. To assess infectious SARS-CoV-2 viral titres, a 50% Tissue Culture Infectious Dose (TCID_50_) assay was performed.

**Reagents**: Australian generic brands of bleach containing 42.6 g/L sodium hypochlorite, vinegar containing 4% *v/v* acetic acid, and dishwashing detergent were purchased from a local supermarket. Sodium hypochlorite (42.6 g/L, Labco) stock solutions were prepared in MilliQ water. Acetic acid stock solutions (4% *v/v*) were prepared in MilliQ water from glacial Acetic acid (>99.7% solution—CAS 64-19-7, Chem-Supply). The 100% ethanol (CAS 63-17-5) was purchased from Chem-Supply. All solutions were filter sterilized then serially diluted for assays as described.

**Cytotoxicity Assay**: Vero cells were plated at 3–5 × 10^5^ cells/well in 96 well plates and allowed to form a monolayer. Cell culture media was then replaced with 25 μL of test-solutions at the concentrations indicated (n = 3/dilution). After 1 h incubation at 37 °C, 5% CO_2_, cells were microscopically examined for morphology, then supernatant removed, washed twice with phosphate buffered saline (PBS), then released from the monolayer with Trypsin-Versine for 5 min at 37 °C, 5% CO_2_. Cells were collected and pelleted at 400× *g* for 3 min, then washed with PBS. After washing, replicate wells were pooled, and cells were stained with the live/dead marker 7AAD (Thermofisher Scientific) according to the manufacturer’s protocols, acquired by flow cytometry on an LSR Fortessa (BD Biosciences) and data analysed using FlowJo v10 software (Becton, Dickinson and Company; 2021).

**Quench Assay**: For the initial set of experiments, we diluted the disinfectants and active ingredients in infection media and assayed them in 10-fold dilutions. For subsequent experiments, we diluted disinfectants and active ingredients in water in 5-fold dilutions. Four independent preparations of each diluted solution were tested for each experiment. A measure of 50 μL of SARS-CoV-2 stocks was added to 50 μL of each diluted solution. After 30 s or 5 min, 900 μL infection media was added to ‘quench’ the disinfection reaction by diluting it 10-fold. Aliquots (25 μL) of each sample were added directly onto confluent Vero cell monolayers (n = 4/dilution) in a 96-well plate and after 30 min at room temperature (18–20 °C) to enable virus adherence, 225 μL infection media containing 1 μg/mL TPCK-trypsin was added and plates returned to the incubator. Three days post-infection (dpi), each well of the plate were examined for monolayer morphology and CPE. Cellular toxicity effects of the disinfectant reagents were also considered, and where CPE could not be assessed due to cell death, samples of supernatants were taken from these wells and passaged again on Vero cells for a further 3 days to investigate for the presence of infectious virus in the absence of chemical cytotoxic effects.

**TCID_50_ reduction Assay**: An equal volume of SARS-CoV-2 of known TCID_50_/mL titre was exposed to each solution diluted in water, resulting in the final concentration as described. In some conditions, where stated, the undiluted stock virus solution was spiked into the test reagents allowing the study of higher concentrations of the household chemicals and enabling a significant reduction in infectious titre to be determined should the chemical be virucidal. After 30 s or 5 min, the virus substrate solution was serially diluted in infection media, and a TCID_50_ assay was performed. Three days post-infection, each well of the TCID_50_ plate were examined for monolayer morphology and virus-induced CPE and the dilution required to infect 50% of cells was determined by the method of Reed and Muench [42].

**Statistics**: Statistical analyses were performed using GraphPad Prism (v9.3).

**Diagrammatic Representation:** Schematics were created with BioRender.com.

## 3. Results

### 3.1. Cell Cytotoxicity Effects of the Household Chemicals Chosen for This Study

To investigate the effectiveness of common household cleaning agents in reducing SARS-CoV-2 infectivity, we selected bleach, detergent, vinegar and alcohol, as well as laboratory-grade sodium hypochlorite and acetic acid, which are the active ingredients for bleach and vinegar, respectively (Table 1). In order to ensure we were measuring the impact of these chemicals on the virus rather than the cells in which it is propagated, we assessed the cytotoxicity of each solution when exposed to Vero cells, the cell line utilized to assess changes to SARS-CoV-2 infectious titres [41,42]. Cell viability was determined using three criteria: (1) whether the cell pellet could be detected after incubation with reagent; (2) cell size/morphology via forward and side scatter profiles detected using flow cytometry; and (3) 7AAD signal detection by flow cytometry due to incorporation of the dye within cells with compromised cellular membranes. We found that Vero cell viability was detrimentally affected when exposed to concentrations greater than 0.1% *v/v* of detergent (Figure 1A), 0.1% *v/v* acetic acid (Figure 1B, which represents a 1/40 dilution of household vinegar); >0.08 g/L bleach and sodium hypochlorite (Figure 1C, equating to a 1/500 dilution of bleach); and >10% *v/v* ethanol (Figure 1D). These results revealed the minimum concentration at which these household chemicals induced cellular toxicity of Vero cells and were taken into account when assessing their impact on the cytopathic effect (CPE) induced by SARS-CoV-2 infection.

### 3.2. Assay Design to Remove Direct Cytotoxic Effects of Household Chemicals on Vero Cells

To investigate whether the household chemicals could be used directly as effective virucidal reagents and render infectious SARS-CoV-2 inert, we undertook two approaches: a ‘50% tissue culture infectious dose (TCID_50_) reduction assay’ and a ‘Quench assay’. For TCID_50_ reduction assays, the goal was to evaluate the concentration of the chemical that ablates or significantly reduces the infectious virus titre compared to that of the untreated inoculum. To do this, a known infectious titre of SARS-CoV-2 was mixed with an equal volume of each chemical prepared at a range of dilutions in water, or a smaller amount ‘spiked’ into solutions to achieve higher concentrations of the household chemicals, when they proved not to be effective at 50% *v/v*. After 30 s or 5 min, samples were taken, and a TCID_50_ assay was performed [42] (Figure 2A). Briefly, a TCID_50_ establishes how dilute the sample containing the infectious virus must be before the virus-induced cytopathic effect is no longer observed in at least 50% of the cells. To evaluate the change in TCID_50_/mL, the cytotoxicity due to the disinfectant remained at the least dilute wells containing the test sample (a mixture of chemical and virus); however, in subsequent dilutions, based on data from Figure 1, cells would not be affected by the disinfectant, and the presence of virus-induced CPE could still be detected, and thus, TCID_50_/mL could be determined and compared to the titre present in the untreated inoculum.

For the Quench assay (Figure 2B), we mixed a known volume of media containing SARS-CoV-2 with an equal volume of each disinfectant diluted in water for 30 s to represent a quick wash or wipe, or for 5 min to represent brief soaking of a surface. The reaction was then quenched with the addition of 10-fold infection media to dilute and to prevent further cytotoxic activity of the reagent, including its impact on the Vero cells based on Figure 1, at the starting dose. Samples were then directly added to washed Vero cell monolayers and observed for SARS-CoV-2-induced CPE, indicating the presence of a replication-competent virus, and recorded as % CPE positive. Thus, the TCID_50_ reduction and the Quench assays are complementary, providing a measure of whether the tested disinfectants were capable of inactivating SARS-CoV-2 and, if so, revealed the lowest effective concentration required to perform this function.

### 3.3. Vinegar Does Not Render SARS-CoV-2 Inactive

Using the TCID_50_ reduction assay, when undiluted vinegar and 4% *v/v* acetic acid was exposed to an equal volume of 10^5.33^ TCID_50_/mL SARS-CoV-2, so that the final concentration of active ingredient was 2% *v/v*, neither solution reduced the infectivity of the virus compared to the untreated virus inoculum (Figure 3A). Likewise, in the quench assay, neither undiluted vinegar nor the equivalent amount of acetic acid revealed virucidal activity at either 30 s or 5 min exposure, as 100% of wells tested positive for CPE (Figure 3B). As a control for the quench assay, virus exposed to water alone (untreated) resulted in 100% of wells testing positive for virus-induced CPE. In order to test higher concentrations, a fixed volume of the virus was spiked into a range of vinegar or acetic acid solutions with final concentrations equivalent to 90, 80, 70, 60% vinegar (equivalent to 3.6, 3.2, 2.8 and 2.4% *v/v* acetic acid respectively, Figure 3C). Surprisingly, these solutions remained largely ineffective at reducing SARS-CoV-2 infectivity, as only 3.6% acetic acid moderately but significantly reduced the infectious virus titre, and only at 5 min, compared to the untreated controls (*p* < 0.01, one-way ANOVA). Thus, despite previous reports of vinegar and acetic acid’s ability to completely inactivate some viruses [43,44], our results indicate that household vinegar is incapable of inactivating SARS-CoV-2, at least not within 5 min of exposure.

While the SARS-CoV-2 inoculum was prepared in HEPES buffered infection media, the dilutions of the household vinegar and acetic acid were in water. When diluted in water, we found the pH of both vinegar and acetic acid to be 2.52 (±0.04 SEM) for concentrations of 1–4% *v/v*. In our tests, we postulated that the presence of HEPES buffer in the virus inoculum may have altered the pH of vinegar or acetic acid in the test sample, potentially limiting the disinfection potential of these agents for the TCID_50_ reduction and quench assays (Figure 3A and Figure 3B respectively). We found that the resulting pH of both the spiked 3.6% *v/v* acetic acid and vinegar solutions were 2.52; therefore, the lack of virucidal activity by vinegar was not a result of pH change.

### 3.4. Ethanol at Concentrations of 40% v/v or Higher Inactivates SARS-CoV-2

The use of alcohol-based hand sanitizers containing more than 60% *v/v* ethanol is a recommended preventative measure against the spread of SARS-CoV-2 [45]. To confirm this and investigate if ethanol is effective at lower concentrations, we tested ethanol over a range of dilutions for its ability to inactivate SARS-CoV-2. Using the quench assay, a 50% *v/v* solution of ethanol was highly effective at inactivating SARS-CoV-2 at both 30 s and 5 min exposure times, but 20% *v/v* ethanol was not (Figure 4A). To investigate a broader range of concentrations, SARS-CoV-2 (10^4.69 ± 0.6^ TCID_50_/mL) was spiked in 10% of the final volume, to ethanol solutions diluted from 100% down to 10%, at 10% *v/v* intervals. These data revealed, with both 30 s and 5 min exposure times, that virus-induced CPE was only detectable in conditions in which ethanol was diluted to 30% *v/v* or lower (Figure 4B). Thus, our results suggest that water-based solutions containing 40% *v/v* ethanol (or above) can prevent SARS-CoV-2 infectivity. However, to ensure a safe margin of error, the recommended dilution of 60–70% ethanol is appropriate.

### 3.5. Household Bleach Is Capable of Abolishing SARS-CoV-2 Infectivity

Bleach is a commonly used disinfectant. We, therefore, investigated a range of dilutions of bleach, and its main active ingredient, sodium hypochlorite, at the same concentration found in bleach (42.6 g/L) on SARS-CoV-2 infectivity. A range of dilutions of bleach, and an equivalent active ingredient concentration of sodium hypochlorite, were prepared in water and, as before, exposed to SARS-CoV-2 and tested in both the quench and TCID_50_ reduction assays. Results taken from both the TCID_50_ and quench assay, reveal that within 30 s of exposure (Figure 5A,B), bleach completely inactivated virus when diluted 1/200 (corresponding to 0.21 g/L of sodium hypochlorite) but not beyond this dilution. For the quench assay, bleach was found to be completely virucidal at the 1/500 dilution after 5 min of exposure, and of the three independent experiments performed, only one well (of the 12 total tested) was positive for the presence of virus-induced CPE for the 30 s exposure. In contrast, only 0.84 g/L sodium hypochlorite solutions, equivalent to 1/50 dilution of bleach, exhibited robust virucidal activity after 5 min exposure in both the TCID_50_ and quench tests, suggesting that other components present in bleach contribute to its virucidal effects.

As a tangent, in our initial testing, we had diluted bleach or equivalent concentration of sodium hypochlorite in our virus infection media (serum-free cell culture media). In these initial quench experiments, complete virus inactivation was only detected upon exposure to 1/4 dilution of bleach (equal to 10.5 g/L sodium hypochlorite) (Appendix A). At further dilutions (equivalent to or below 1/40 bleach or 1.05 g/L sodium hypochlorite), no antiviral activity was detected at 30 s, while only partial activity was seen after 5 min of exposure (Appendix A). To determine if this was an effect of the infection media on the activity of the sodium hypochlorite, we then compared side by side the antiviral activity of dilutions of bleach and sodium hypochlorite prepared in infection media or water (Figure 6). Our results demonstrate that bleach and sodium hypochlorite were less effective when diluted in infection media compared to when diluted in water. Given the presence of HEPES buffer in the infection media, we postulated that the buffer may have impacted the pH of the test solution and may offer a reason why the bleach and sodium hypochlorite solutions were less effective as virucidal agents when prepared in this media. We found a marked decrease in pH in solutions containing lower amounts of bleach or sodium hypochlorite when diluted in infection media, compared to when the agents were diluted in water (Figure 6C). This may be the reason why bleach and sodium hypochlorite are less virucidal when diluted in HEPES buffered media compared to when diluted in water. Therefore, caution should be used if diluting bleach in agents other than water as they may impact its virucidal activity against SARS-CoV-2, particularly if it is diluted in buffered solutions such as culture media which may be relevant for disinfection of laboratory material.

### 3.6. Dishwashing Detergent Diluted up to 500-Fold Inactivates SARS-CoV-2

As above, we also tested a range of dilutions of generic dishwashing detergent for its ability to inactivate SARS-CoV-2. Given the cytotoxicity data (Figure 1), we created a stock solution of detergent in water at a 10% *v/v* concentration, then tested a series of dilutions for antiviral activity against SARS-CoV-2. Using the TCID_50_ reduction assay, we found that direct exposure of SARS-CoV-2 to solutions of detergent in water, ranging in final concentrations from 5% to 0.2% *v/v*, abolished the ability to detect infectious virus within 30 s (data not shown) that was maintained at 5 min (Figure 7A). Exposure of virus to 0.04% *v/v* detergent for 5 min did not significantly reduce SARS-CoV-2 titre compared to virus exposed to water alone (Figure 7A). Similarly, using the quench assay, we found that dilutions as low as 0.2% *v/v* detergent completely inactivated SARS-CoV-2 at 30 s or 5 min, while further dilution resulted in incomplete inactivation (Figure 6). Taken together, these assays suggest that detergent is virucidal at 0.2% *v/v* dilution in water, which is equivalent to a 1/500 dilution of detergent from the original solution.

### 3.7. Combining Detergent with Bleach Does Not Significantly Enhance Virucidal Activity

Given that cleaning products are sometimes combined in order to achieve synergistic or additive virucidal effectiveness [46], we tested whether the combination of bleach and detergent could have additive antiviral activities against SARS-CoV-2. When the combined dilutions of these agents were prepared in water and examined for antiviral activity in the quench assay, no cumulative effect was observed, as the minimum effective doses of bleach and detergent against SARS-CoV-2 were similar to the activity of detergent alone (Figure 8A,B).

### 3.8. SARS-CoV-2 Remains Infectious through Multiple Freeze-Thaw Cycles

We and others have previously shown that SARS-CoV-2 is more stable at lower temperatures, retaining a high level of infectious activity after 2 weeks at 4 °C while losing activity by 1–3 days at 37 °C [13,47]. There have also been reports of SARS-CoV-2 genome presence and potential to culture infectious virus recovered from the surface of frozen packaging [18,48]. Given cold-chain transportation in the frozen food industry prior to reaching the consumer can result in multiple freezing and thawing cycles, we tested whether SARS-CoV-2 would remain infectious if it underwent such conditions and thus remain a potential infectious fomite source. To do this, we subjected the virus to freeze-thaw cycles every 24 h and assayed for infectious titre each time (Figure 9). Up to 7 freeze-thaw cycles were tested, and remarkably, we found no significant changes in infectious titre (one-way ANOVA). These data highlight the potential for infectious virus to be present on fomites such as food packaging that has been kept cold or frozen for long periods.

## 4. Discussion

To limit the transmission potential of SARS-CoV-2, many health authorities and governmental agencies have mandated isolation when infected, mask wearing in public, and stressed the importance of hand hygiene [49]. In addition to these behavioural changes, the evidence provided by some studies that infectious SARS-CoV-2 can persist on some surfaces [11,12,13] has meant many global authorities commit to temporary closure and deep cleaning of exposure sites in order to limit potential fomite transmission [36,37,38]. While the potential risk of transmission by direct contact with contaminated surfaces has been largely debated for SARS-CoV-2, particularly because aerosolized transmission can never be fully ruled out [18,19,20,21,22,23,24,25,26], there remains a risk of cluster outbreaks due to fomite spread, perpetuating the pandemic [22,48].

A role for contaminated surfaces in the transmission of viruses was first proposed half a century ago [50], with subsequent research supporting this as a likely route for transmission of respiratory viruses [51]. As an example, a study based on an influenza A virus fomite contamination of a surface has shown that a 5 s contact exposure can transfer up to 31.6% of the droplet infectious viral load [52], which could then be easily transferred to a permissible infection site on the body. Previous studies have suggested that viruses, including SARS-CoV-2 and influenza, can remain infectious on surfaces for extended periods of days to months [11,13,53,54]. Highlighting the need for evidence-based research on the effectiveness of household chemicals to decontaminate surfaces potentially exposed to SARS-CoV-2, a published survey conducted by the CDC revealed that more than 80% of respondents did not feel they knew how to disinfect their home safely to prevent SARS-CoV-2 transmission, despite increasing their household cleaning efforts [39].

To clean a surface, typically, either a solution is sprayed onto the site, or a cloth is wet with the cleaning solution, then the surface is wiped, both of which are methods known as carrier-based decontamination. Our study did not test these methods directly, but instead tested the virucidal activity of the household chemicals when in direct contact with the virus in solution in a minimal volume, similar to that encountered during carrier-based decontamination of surfaces. Our study investigated the antiviral activity of several household generic surface cleaning products and found that bleach (up to 1/200 dilution), dishwashing detergent (up to 1/500 dilution) and alcohol-containing solutions (at least 40% *v/v*), but not vinegar, was effective at inactivating SARS-CoV-2. We also tested the stability of SARS-CoV-2 infectivity over multiple freeze-thaw cycles and found that infectious virus was remarkably stable and was not affected by up to seven freeze-thaw cycles.

Our findings reveal that vinegar, even when used almost neat from the bottle (90% *v/v*), or the equivalent amount of its active ingredient (3.6% *v/v* acetic acid in water) is almost completely ineffective at inactivating SARS-CoV-2, and, therefore, should not be used to disinfect surfaces suspected to be contaminated with this virus. This observation contrasts with previous studies reporting that 10% *v/v* dilution of vinegar (equivalent to 0.4% *v/v* acetic acid) can inactivate influenza A virus subtypes H7N2 [55] and H1N1 [56]. Whilst vinegar fails to degrade the influenza viral RNA genome [56], the acetic acid present in vinegar has been shown to induce conformational changes of the viral surface glycoprotein, hemagglutinin (HA) that is involved with cellular attachment and infection of the host and thus limits the ability for the virus to infect [57]. Our findings suggest that SARS-CoV-2 surface Spike glycoprotein (essential for attachment and infection of a cell) is not susceptible to this influence of acetic acid or in a solution with acidic pH. The pH of the vinegar we tested was 2.5, and our findings are in line with, and extend upon, a published study that the infectivity of SARS-CoV-2 is not significantly altered when the acidity of the media containing the virus is reduced to as low as pH = 3 [13].

Alcohol, and alcohol-based hand sanitizers, are recommended by the WHO, CDC and several national governments for use to decontaminate hands for protection against microbial agents and viruses, including SARS-CoV-2, other coronaviruses and influenza viruses [40,45]. Our study confirms that ethanol in solution can inactivate SARS-CoV-2, even at concentrations as low as 40%. As methylated spirits, a household cleaning agent accessible to the public in supermarkets contains 95% ethanol, we expect the effective virucidal concentration to remain similar to laboratory-grade ethanol. An important consideration is that ethanol-containing wipes or handwashes, which typically contain more than 60% *v/v* alcohol, may become ineffective if the user dilutes the overall solution to below 40% *v/v* alcohol or if the product has been improperly stored and alcohol evaporated over time.

We found dilute bleach to be effective at rendering SARS-CoV-2 non-infectious at a maximum of 1/200 dilution of the solution in water within 30 s of exposure. Similar, albeit less potent, activity was observed for the equivalent amount of the active ingredient in bleach, sodium hypochlorite, which was fully active at dilutions as low as 0.21 g/L. Thus, providing that bleach is prepared in water at concentrations above this, exposure to potentially contaminated surfaces should rapidly inactivate SARS-CoV-2. Our findings were congruent with those of Chin et al., who found that a 1:99 dilution of bleach was capable of inactivating SARS-CoV-2 [13]. Most commonly, bleach and sodium hypochlorite solutions are made up in water for use as disinfecting agents. However, it is also common for virology laboratories to use bleach traps and bleach bottles for capturing and inactivating spent culture medium containing virus. Given that our results show buffered media may adversely impact the inactivating effects of the bleach, laboratories should be cautious not to dilute the solution beyond 10.5 g/L sodium hypochlorite to remain effective against SARS-CoV-2.

Our study also shows that household dishwashing detergent was an effective way to render SARS-CoV-2 non-infectious, as a 1 in 500-fold dilution in water effectively inactivated the virus. Congruent with our findings, some detergents have been found to be more effective than alcohol containing handwashes for other pathogens, including rhinoviruses, which are causative agents of the common cold [58] and bacteria [59].

Importantly, in a published CDC survey, 25% of respondents reported at least one adverse health effect from inappropriate use of household cleaning agents and disinfectants, with those who were engaging in risky practises, such as mixing chemicals, more frequently reporting adverse health effects [39]. Antiseptic and household cleaning products available in the supermarket routinely contain combined surfactants (detergents), alcohols, acids and antimicrobial chemicals. We also tested a combination of bleach and detergent but found the combination did not lead to a marked increase in the virucidal activity of the detergent-solution alone against SARS-CoV-2. Furthermore, as we observed with bleach diluted in culture media, combining different products containing different active ingredients and buffers may reduce the antiviral activity of the products compared to when used independently. Mixing of chemicals may also pose a significant health risk for the user due to the potential production of noxious gases and an increase in toxicity [39].

Lastly, several studies have suggested that the persistence of coronaviruses on surfaces is reduced at temperatures higher than 37 °C when compared to room temperature (20 °C) [13,60]. Our published study on SARS-CoV-2 infectivity when stored in virus transport media at 37 °C also revealed a significant reduction in titre within 48 h and, when stored at 45 °C, completely inactivates the virus in less than 16 h [47]. Reports of potential transmission of infectious SARS-CoV-2 from frozen packaging [18,48] led us to investigate the stability of the virus when exposed to multiple freeze-thaw cycles, and our findings revealed that this virus is extremely resistant to freeze-thaw mediated damage. Thus, even when virus transmission has been eliminated in a community, the virus may remain viable if it is present in the correct environment, such as on a surface of a product kept in refrigerated cold storage or in a freezer and on frozen goods shipped into that community.

## 5. Conclusions

In conclusion, our study provides experimental evidence for the concentrations of common household chemicals that are needed for the effective inactivation of SARS-CoV-2. Surprisingly, while vinegar has been shown to be effective against influenza A virus, our study reveals that vinegar is unable to render SARS-CoV-2 inactive, and we do not recommend its use for this purpose. Our study also explored several other household cleaning solutions that are effective against SARS-CoV-2. This includes bleach at a maximum of 200-fold dilution in water (equivalent to 0.21 g/L of sodium hypochlorite), as well as dishwashing detergent, diluted up to 500-fold in water and ethanol diluted as low as 40% *v/v* (Table 2). There was no reduction in virus activity after multiple freeze-thaw cycles, highlighting the need to consider that virus may survive for long periods on surfaces in the right conditions and may theoretically lead to new infections even in the absence of recent community spread.

## Figures and Tables

**Figure 1 viruses-14-00715-f001:**
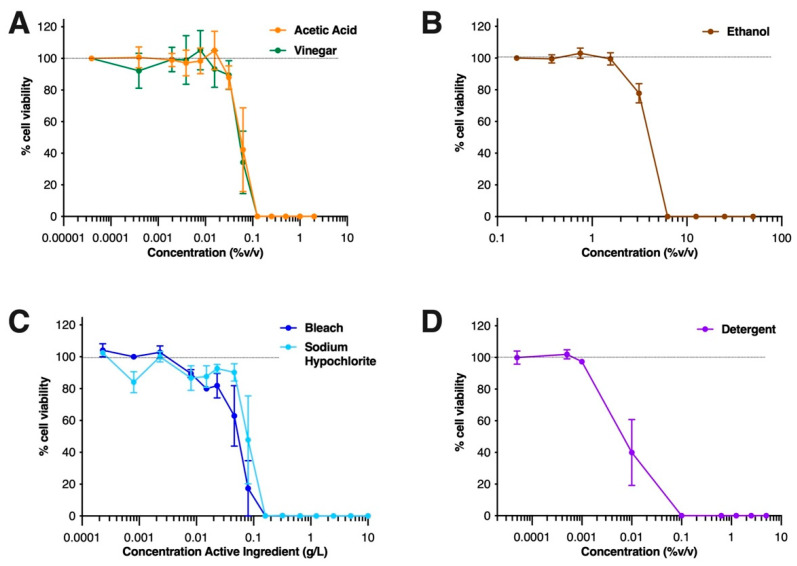
**Cytotoxicity of household chemicals on Vero cells:** Vero cells were exposed to (**A**) vinegar or its active component, acetic acid, (**B**) ethanol, (**C**) bleach, or its active component, sodium hypochlorite, or (**D**) detergent, at various concentrations for 1 h at 37 °C, 5% CO_2_. Cells were microscopically examined for morphology, and after removal of the test-solution, replicate dilutions were pooled and stained with the live/dead marker 7AAD by flow cytometry within 1 h. Graphs show mean % viable cells ±SEM of data pooled from four independent experiments. The dotted line represents cell viability when in media alone (i.e., not exposed to any chemical).

**Figure 2 viruses-14-00715-f002:**
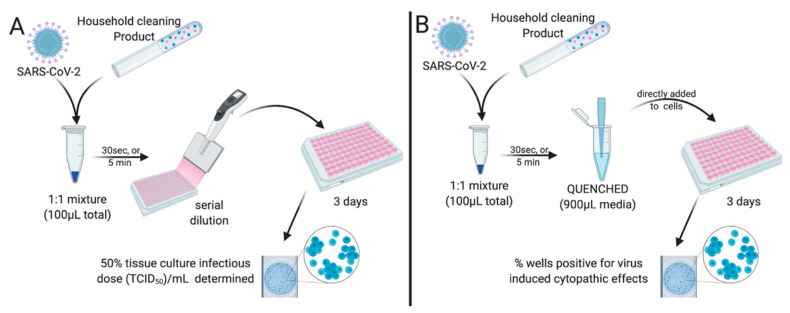
**TCID_50_ reduction and Quench assays.** Diagrammatic representation of workflow for (**A**) TCID_50_ reduction assay and (**B**) Quench Assay. Briefly, at room temperature (18–20 °C, 40% relative humidity), SARS-CoV-2 in suspension was mixed with an equal volume of test solution containing known concentration of active ingredients for 30 s or 5 min. Serial dilutions of the active ingredients were tested in order to determine the lowest concentration of active ingredient required for virus neutralisation. Following this, either: (**A**) a TCID_50_ was performed (n = 4 replicates/test dilution), or (**B**) the reaction was quenched with the addition of 900 μL infection media in order to prevent further activity at the starting dose. Samples (n = 4 replicates/test dilution) were then directly added to washed Vero cell monolayers and 3 days later, observed for SARS-CoV-2-induced CPE.

**Figure 3 viruses-14-00715-f003:**
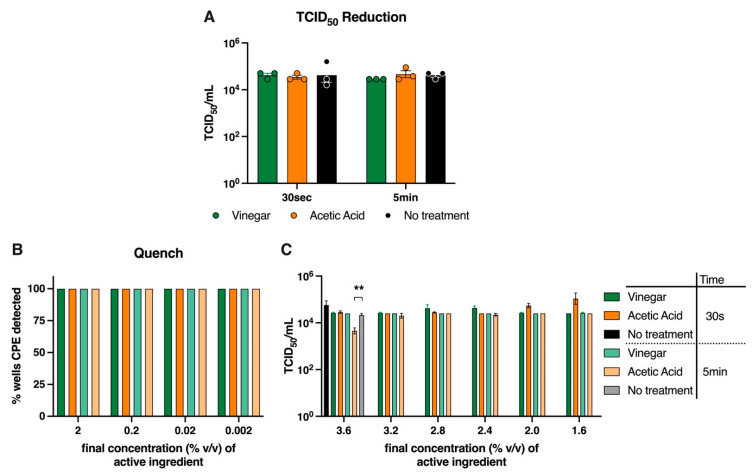
**Vinegar is incapable of rendering SARS-CoV-2 non-infectious.** At room temperature (18–20 °C, 40% relative humidity), exposure of SARS-CoV-2 to vinegar or acetic acid for 30 s or 5 min prior to detection of infectious virus by the (**A**) TCID_50_ reduction assay, (n = 3 replicates/test dilution) or (**B**) Quench assay (n = 4 replicates/test dilution). (**C**) Vinegar and acetic acid solutions (n = 3 replicates/test dilution), at the concentrations indicated, were spiked with 10^4.54 ± 0.3^ TCID_50_/mL SARS-CoV-2 and 30 s or 5 min later assayed for infectious virus titre. Graphs show mean ± SEM of data pooled from three independent experiments. ** *p* < 0.01 acetic acid (5 min) vs. no treatment (5 min), one-way ANOVA. Key applicable to both (**B**) and (**C**).

**Figure 4 viruses-14-00715-f004:**
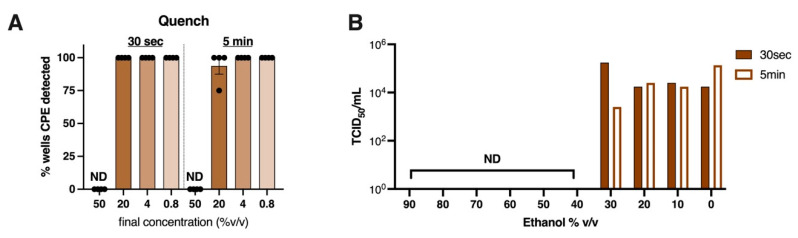
**Ethanol renders SARS-CoV-2 non-infectious when used at more than 40% *v/v*:** (**A**) % wells positive for virus induced cytopathic effect after SARS-CoV-2 was exposed to a range of dilutions of ethanol for 30 s or 5 min using the Quench assay. Graph shows mean + SEM of data pooled from four independent experiments. (**B**) A fixed amount of SARS-CoV-2 (10^4.69 ± 0.6^ TCID_50_/mL) was spiked into a range of diluted ethanol solutions and 30 s or 5 min later assayed for remaining infectious virus titre by TCID_50_. Experiments were performed at room temperature (18–20 °C, 40% relative humidity). Graph shows mean ± SEM of data representative of 3 independent experiments. ND = virus induced CPE was not detectable.

**Figure 5 viruses-14-00715-f005:**
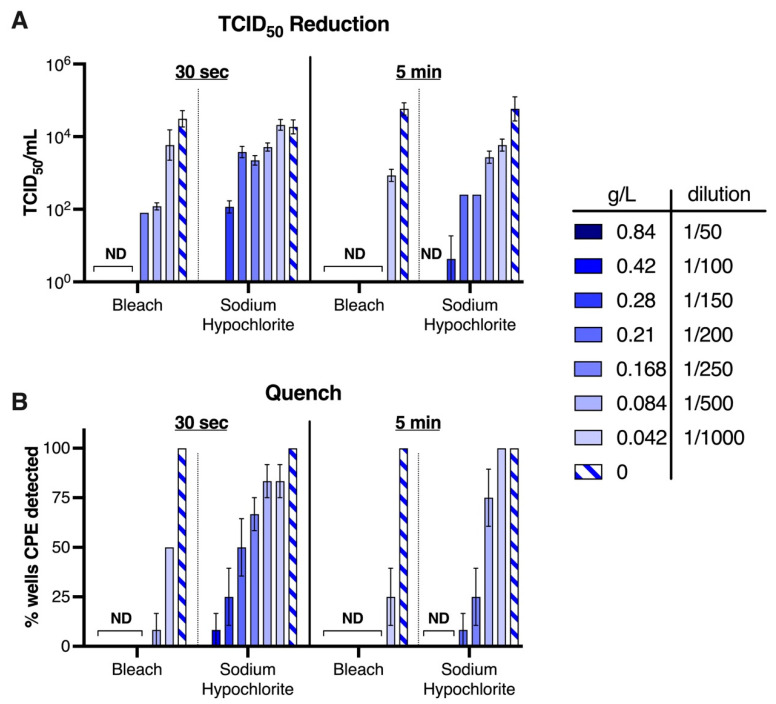
**Household bleach can render SARS-CoV-2 inactive.** Exposure of SARS-CoV-2 to bleach or sodium hypochlorite (n = 4 replicates per dilution) for 30 s or 5 min revealed a concentration dependent reduction in infectious virus titre by the (**A**) TCID_50_ reduction assay, or (**B**) Quench assay. Experiments were performed at room temperature (18–20 °C, 40% relative humidity). Graphs shows mean ± SEM of collated data from three independent experiments. ND = virus induced CPE was not detectable.

**Figure 6 viruses-14-00715-f006:**
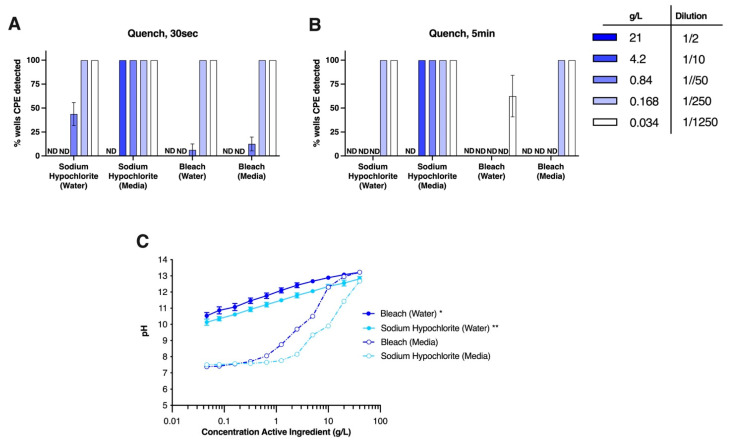
**Preparation of bleach in infection media adversely affects the virucidal concentration.** Using the quench assay to test virucidal activity, sodium hypochlorite and bleach were diluted in either infection media or water, then exposed to SARS-CoV-2 for (**A**) 30 s or (**B**) 5 min. ND = virus-induced CPE was not detectable. (**C**) pH of bleach and sodium hypochlorite solutions diluted in either media or water. * Bleach: *p* < 0.05 when diluted in water compared to media at final concentration 10 g/L and *p* < 0.001 water compared to media at concentrations less than 10 g/L. ** Sodium hypochlorite: *p* < 0.001 when diluted water compared to media from 0.046–20 g/L. Two-way ANOVA with Tukey’s multiple comparisons test. Experiments were performed at room temperature (18–20 °C, 40% relative humidity). Graphs show mean ± SEM of data pooled from four independent experiments for (**A**) and (**B**) and 2 experiments for (**C**).

**Figure 7 viruses-14-00715-f007:**
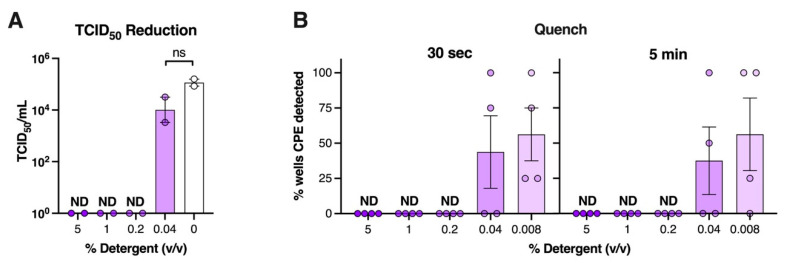
**A 1 in 500 dilution of dishwashing detergent rendered SARS-CoV-2 non-infectious.** (**A**) TCID_50_/mL of SARS-CoV-2 after 5 min exposure to dishwashing detergent diluted in water (n = 4 replicates/test dilution), Graphs show mean ± SEM of data pooled from two independent experiments. ns = data not significant, one-way ANOVA. (**B**) % wells positive for virus induced cytopathic effect after SARS-CoV-2 was exposed to a range of dilutions of detergent for 30 s or 5 min using the quench assay. Experiments were performed at room temperature (18–20 °C, 40% relative humidity). ND = CPE was not detectable. The more concentrated the solution (% given in x-axis below), the darker the colour.

**Figure 8 viruses-14-00715-f008:**
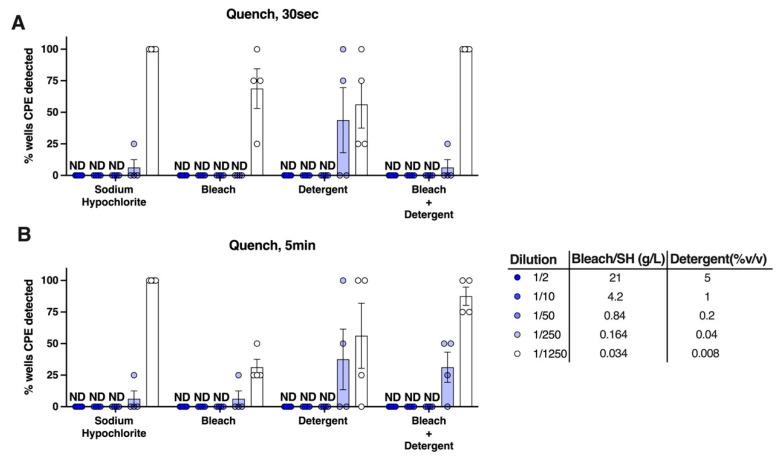
**Combining detergent with bleach does not increase efficacy of SARS-CoV-2 inactivation.** Using the quench assay to test virucidal activity, detergent was added to bleach or sodium hypochlorite (SH) diluted in water then then exposed to SARS-CoV-2 for (**A**) 30 s or (**B**) 5 min. Experiments were performed at room temperature (18–20 °C, 40% relative humidity). Graphs show mean ± SEM of data pooled from four independent experiments. ND = virus-induced cytopathic effect was not detectable.

**Figure 9 viruses-14-00715-f009:**
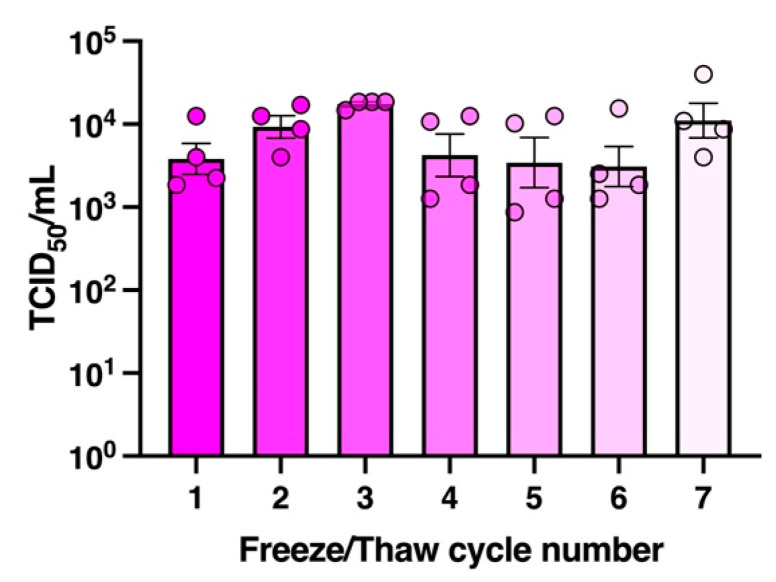
**SARS-CoV-2 remains infectious after multiple freeze-thaw cycles.** Four aliquots of SARS-CoV-2 stocks (10^5.24^ TCID_50_/mL) stored at −80 °C, were thawed to room temperature (18–20 °C) (cycle 1) and assayed for infectious virus titre, then the remaining aliquot re-frozen at −20 °C. The freeze-thaw cycle was repeated every 24 h for 6 days and infectious titre determined by TCID_50_ upon each thaw. Data not significantly different (one-way ANOVA). Graph shows mean ± SEM.

**Table 1 viruses-14-00715-t001:** Household cleaning solutions and their active ingredient.

Household Cleaning Solution	Active Ingredient
Vinegar	4% *v/v* Acetic acid
Alcohol (e.g., Methylated spirits, hand sanitizers)	Ethanol
Bleach	42.6 g/L Sodium hypochlorite
Dishwashing detergent	unknown *

* reported to contain <10% sodium laureth sulfate and <10% Lauramidopropo-dimethylamineoxide.

**Table 2 viruses-14-00715-t002:** Summary of effective dilutions (in water) of household cleaning solutions capable of rendering SARS-CoV-2 inactive.

Household Cleaning Solution	Active Ingredient	Effective Dilution in Water ^Δ^
Vinegar	4% *v/v* Acetic acid	Not virucidal against SARS-CoV-2
Alcohol (e.g., Methylated spirits, hand sanitizers)	Ethanol	40% *v/v*
Bleach	42.6 g/L Sodium hypochlorite	1 in 200; equivalent to 0.21 g/L Sodium hypochlorite
Dishwashing detergent	unknown *	1 in 500
Freeze (−20 °C)/Thaw cycles	Not applicable	Not virucidal against SARS-CoV-2

* reported to contain <10% sodium laureth sulfate and <10% Lauramidopropo-dimethylamineoxide. **^Δ^** We found these final concentrations of cleaning solution to be effective within 30 s of exposure to virus.

## Data Availability

Please contact the corresponding author for data supporting results generated for this study.

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
