# Peer review of "The Efficacy of Common Household Cleaning Agents for SARS-CoV-2 Infection Control"

_viruses, 2022, doi:10.3390/v14040715_

Round 1

Reviewer 1 Report

The manuscript by Almeida et al. is a nicely written report of efficacy testing for various microbicidal actives against SARS-CoV-2. I especially appreciate the evaluation of various concentrations of the test agents, providing valuable information on the concentration-dependence of the efficacy when applied at the relevant contact times of 0.5 and 5 min. The freeze/thaw data are also useful.

Suggested title change: The efficacy of common household cleaning agents for SARS-CoV-2 infection control

In general, virucidal efficacy is dependent not only on the virucide and its concentration but also the contact time and the temperature at which inactivation is measured. You have done a good job of emphasizing contact time and concentration, but less so for temperature. I advise that this critical parameter be called out in the abstract and figure legends so that readers do not have to hunt in the method section for this information.

Also, though the general theme of the paper seems to be around surface cleaning (final line of abstract, final sentence of introduction), your methodology was not a carrier based test but a solution inactivation test. Do you think this deserves an explanation? If so, please consider addressing  this in the discussion section as a possible limitation of the study design.

Abstract, final sentence, suggest: “These results should help inform readers as to how to effectively disinfect surfaces and objects that have potentially been contaminated with SARS-CoV-2, using common household chemicals.”

Introduction, line 32, suggest: “This continued transmission has meant host-adaptive evolution of the virus leading to isolates of increased transmissibility, virulence and pathogenicity (coined variants of concern (VOC)).”

Introduction, line 52: The term “antiseptics” is usually reserved for agents applied to wounds. Suggest the term microbicides or virucides (this applies also to the use of “antiseptics” on line 165).  The citations used here (29-32) seem an odd choice. If you are discussing surface hygiene, references 30 and 32 may not be applicable. On the other hand, there are several reviews of microbicidal efficacy against SARS-CoV-2 on surfaces, and some primary papers as well. For instance:

Gerlach M, Wolff S, Ludwig S, Schäfer W, Keiner B, Roth NJ, et al. Rapid SARS-CoV-2 inactivation by commonly available chemicals on inanimate surfaces. Journal of Hospital Infection. 2020;106:633-634.

Ijaz MK, Nims RW, Zhou SS, Whitehead K, Srinivasan V, Kapes T, et al. Microbicidal actives with virucidal efficacy against SARS-CoV-2 and other beta- and alpha-coronaviruses and implications for future emerging coronaviruses and other enveloped viruses. Scientific Reports. 2021;11.5626. https://doi.org/10.1038/s41598-021-84842-1

 Jahromi R, Mogharab V, Jahromi H, Avazpour A. Synergistic effects of anionic surfactants on coronavirus (SARS-CoV-2) virucidal efficiency of sanitizing fluids to fight COVID-19. Food and Chemical Toxicology. 2020;145:111702.

Schrank CL, Minbiole KPC, Wuest WM. Are quaternary ammonium compounds, the workhorse disinfectants, effective against severe acute respiratory syndrome coronavirus-2. ACS Infectious Diseases. 2020;6:1553-1557.

Kwok CS, Dashti M, Tafuro J, Nasiri M, Muntean E-A, Wong N, et al. Methods to disinfect and decontaminate SARS-CoV-2: a systematic review of in vitro studies. Therapeutic Advances in Infectious Disease. 2021;8:1-12. DOI: 10.1177/2049936121998548

Results page 7, line 268 .suggest: “... bleach completely inactivated virus when diluted 1/200 (corresponding to 0.21 g/L of sodium ...”

Results, page 8, line 302. It might be worth noting here that most commonly, bleach and sodium hypochlorite solutions are made up in water for use as disinfecting agents. However, as you state, it is also common for virology laboratories to use bleach traps and bleach bottles for capturing and inactivating spent tissue culture medium containing virus. In that case, your result that the buffered media may adversely impact the inactivating effects of the bleach is useful to keep in mind.

Results, page 10, lines 338-340, suggest: “Given that cleaning products are sometimes combined in order to achieve synergistic or additive virucidal effectiveness [43], we tested whether the combination of bleach and detergent could have additive antiviral activities against SARS-CoV-2.”

Reviewer 2 Report

Very interesting work, the authors did a great job to test the impact of common household cleaning agents on SARS2 control.

Minor comments:

  1. line 121, either in the method 'TCID50 reduction assay' or in the corresponding result section, need to specify the dilution factor
  2. line 131, the statistical method needs to be more specific
  3. line 157, please add the control (blank) group, in which no household chemicals were added. for any experiment to support a scientific point, the control is needed
  1. line 231, for this figure and the rest ones, suggest to modify the color coding, or, it's easy to makes confusion
